# High-Power Ultrasound and High-Voltage Electrical Discharge-Assisted Extractions of Bioactive Compounds from Sugar Beet (*Beta vulgaris* L.) Waste: Electron Spin Resonance and Optical Emission Spectroscopy Analysis

**DOI:** 10.3390/molecules30040796

**Published:** 2025-02-09

**Authors:** Josipa Dukić, Anet Režek Jambrak, Jurica Jurec, Dalibor Merunka, Srećko Valić, Rafaela Radičić, Nikša Krstulović, Marinela Nutrizio, Igor Dubrović

**Affiliations:** 1Faculty of Food Technology and Biotechnology, University of Zagreb, 10000 Zagreb, Croatia; josipa.dukic@pbf.unizg.hr (J.D.); marinela.nutrizio@pbf.unizg.hr (M.N.); 2Ruđer Bošković Institute, 10000 Zagreb, Croatia; jjurec@irb.hr (J.J.); dalibor.merunka@irb.hr (D.M.); srecko.valic@irb.hr (S.V.); 3Department of Medical Chemistry, Biochemistry and Clinical Chemistry, Faculty of Medicine, University of Rijeka, 51000 Rijeka, Croatia; 4Centre for Micro- and Nanosciences and Technologies, University of Rijeka, 51000 Rijeka, Croatia; 5Institute of Physics, Bijenička Cesta 46, 10000 Zagreb, Croatia; rradicic@ifs.hr (R.R.); niksak@ifs.hr (N.K.); 6Department of Environmental Health, Teaching Institute of Public Health of Primorje-Gorski Kotar County, 51000 Rijeka, Croatia; igor.dubrovic@zzjzpgz.hr

**Keywords:** high-voltage electrical discharge, high-power ultrasound, sugar beet leaves, bioactive compounds, antioxidant capacity, electron spin resonance spectroscopy, optical emission spectroscopy

## Abstract

To achieve sustainable extractions, this study examines the impact of different extraction methods to utilize waste from the sugar industry. In addition to conventional thermal extraction, the impact of high-power ultrasound (US) and high-voltage electrical discharge (HVED)-assisted extractions on the yield of bioactive compounds and the antioxidant capacity (AC) value of sugar beet leaf extracts was determined. US extraction proved to be an excellent method for extracting bioactive compounds, while HVED extraction proved to be an excellent method for extracting Vitexin. AC was measured both spectrophotometrically (DPPH and FRAP) and spectroscopically via electron spin resonance (ESR). The AC results correlate with each other, and the highest AC values were found in the US-treated samples with 25% ethanol solution as the extraction solvent. Characterization of the plasma via optical emission spectroscopy (OES) showed that neither the solvent nor the sample influenced the plasma spectra, only the gas used (nitrogen/argon). All of the obtained results provide an excellent basis for future research into the utilization of food waste and by-products.

## 1. Introduction

In the face of global challenges such as resource depletion, environmental degradation, and the increasing food needs of a rapidly growing population, the principles of sustainable development are critical to modern food production and the stability of the food industry. At its core, sustainable development stands for responsible, intelligent and creative use of natural resources, for social policies that are fair to all people regardless of age, ethnicity and gender, and for long-term economic stability based on economic growth without negative impacts on nature and the population. By accepting the Sustainable Development Goals (SDGs) and incorporating the defined practices, the opportunity is created to protect fragile ecosystems, reduce greenhouse gas emissions and preserve valuable resources for future generations. The concept of sustainable development is central to the future of food production, considering that with current trends, the population is expected to increase rapidly, most likely exceeding 9 billion people by early 2050, according to some estimates [1]. In the face of population growth, the food sector is also facing the severe consequences of climate change, leading to losses of raw materials for production and thus jeopardizing the security of the supply chain. Global food production is expected to increase by 70%, leading to increased demand for raw materials and energy [2]. Therefore, innovations in the food sector are increasingly focusing on new, more energy-efficient and environmentally friendly raw materials and processing methods, which are recognized as valuable tools to achieve the SDGs [3,4,5,6].

In modern food technology and production, the use of ultrasound (US) represents an extremely valuable technology that, in addition to its versatility, has enabled an increase in processing efficiency, an improvement in product quality and a reduction in energy consumption. Regardless of whether it is high- or low-intensity US, this technology is widely used in the food industry for quality control, production, hygiene and by-product/waste processing/extraction [7]. Previous studies have shown that some of the advantages of using US in the extraction of bioactive components are the reduced extraction time without loss of yield, the low process temperature which reduces the degradation of sensitive compounds, the lower energy and solvent consumption, the possibility of using water as a solvent and thus avoiding toxic waste, the possibility of applications in the extraction of different bioactive compounds, the simplicity of the process and the facilitated subsequent separation and purification of the extracted components [8]. The above-mentioned advantages of US are due to the mechanical and thermal effects of transient acoustic cavitation, which cause the destruction of cell membranes, increased mass transport and more thorough penetration of solvents into the intracellular matrix [9].

Due to the numerous shortcomings of traditional, conventional methods for the extraction of bioactive compounds, innovative “green” non-thermal extraction methods that are economical and environmentally friendly are increasingly being used. The use of non-thermal extraction methods reduces energy and solvent consumption, increases the efficiency and selectivity of the process, ensures yields are higher, and makes automation of the process possible. In addition to US-assisted extraction, non-thermal techniques also include high-voltage electrical discharge (HVED)-assisted extraction, a non-thermal technique in which plasma is generated. Plasma is one of the four fundamental states of matter, i.e., a fully or partially ionized gas consisting of cations, anions, radicals, electrons and excited and unexcited atoms [10]. Some of the practical applications of this method are water purification, inactivation of microorganisms (bacteria, viruses, yeasts), electrohydraulic comminution and extraction of bioactive compounds from plants [11]. There is a difference between thermal and non-thermal, i.e., cold plasmas. In thermal plasmas, the particles (electrons, ions and neutrons) are in thermal equilibrium, i.e., they have the same temperature. In cold plasmas, electrons have different electron and ion temperatures and are not in local thermal equilibrium with ions and neutrons. Electrons can collide with other present molecules (N_2_, O_2_ and H_2_O), creating secondary electrons, photons, ions and radicals in the plasma [12]. Non-thermal cold plasma generated by corona and current discharges implies the creation of plasma without heating the medium. Although the temperature in the discharge channels can be very high, a moderate or room temperature can be maintained due to the small volume and local nature of the discharge channels [13].

The low temperature of the extractions makes these methods desirable for the extraction of bioactive components with antioxidant capacity. Antioxidants are molecules that are crucial for the proper functioning of cells and biochemical processes, as they provide protection against oxidative stress and negative reactions with free radicals. Phenolic compounds are a large group (flavonoids, phenolic acid, tannins, stilbenes and lignans) of natural antioxidants found in plants. They are usually biosynthesized via shikimic acid from phenylalanine or tyrosine as initial reactants, and the hydroxyl (OH) group in the benzene ring is responsible for their antioxidant capacity [14]. In addition to the presence, the position and number of OH groups are also of great importance for the biological activity of flavonoids [15]. Phenolic compounds belong to the group of primary antioxidants, which are characterized by high efficacy and easy regenerability [16]. The consumption of phenolic compounds from food has been shown to have positive effects on human health, such as anti-inflammatory, anti-cancer and antiviral effects; improvement of immune system function; prevention of gastrointestinal tract diseases; etc. [17]. In the face of modern challenges such as climate change, the increase in external stressors and the huge amounts of food waste, scientific research in the food industry has focused on the use of advanced green processing techniques in the extraction of phenolic compounds from already existing but equally valuable waste such as sugar beet leaves [18,19]. Considering that sugar beet leaves represent about 30% of the whole plant, it is important to consider the use of this waste as a potential raw material for sustainable production needs [20]. The beneficial nutrient profile of leaves does not only refer to the macronutrients they contain. In terms of micronutrients, plant leaves, including sugar beet leaves, are generally a good source of minerals (Ca, Cu, Fe, K, Mg, Mn), vitamins (A, C, E, K) and phytochemicals/secondary metabolites (chlorophylls, carotenes), some of which have been shown to have a positive impact on human health [20,21]. There are relatively few studies on the extraction and characterization of the bioactive components contained in sugar beet leaves, but in general, sugar beet leaves contain high amounts of gallic acid (344 µg/mL) and ferulic acid (89.7 µg/mL), while cinnamic and caffeic acids and catechin are present in small amounts [22]. In addition, vitexin was found to be the most abundant phenolic compound [23].

For the above reasons, this study evaluated the efficacy of US and HVED treatments on the extraction of bioactive components with antioxidant capacity from sugar beet leaves by varying the extraction solvent, applied amplitude and treatment time for US treatments and the extraction solvent, applied gas, applied voltage and treatment time for HVED treatments. The focus is on sustainability and the use of industrial waste as a source of bioactive compounds combined with a low-environmental-impact extraction method.

## 2. Results and Discussion

### 2.1. Total Phenolic Content (TPC)

The results of the TPC yield for US-treated samples with deionized water as an extraction solvent were already published by Dukić et al. [19], but the extractions were repeated due to the second batch of plant material, and the results of the TPC yield for all US-treated samples are presented graphically (Figure 1a). The trend of previously published results is the same now, with a slight decrease in TPC yield observed in the new batch of plant material.

The reason for this decrease is due to the storage time and storage conditions of the plant material, which directly affect the chemical composition of the leaf itself. As in the previously mentioned work, the highest yield of TPC (16.47 ± 0.10 mg/g_d.m._) was determined to be that of the 0LU6 sample (with an amplitude of 100%, a treatment time of 9 min and deionized water as the extraction solvent). Maravić et al., 2022, reported a similar TPC yield for US-treated sugar beet leaf samples [23]. The increase in ethanol content (in the extraction solvent) directly affected the decrease in TPC yield in the US-treated samples, and thus the lowest TPC yield of 5.85 ± 0.21 mg/g_d.m._ was recorded for the 50LU7 sample (with an amplitude of 50%, a treatment time of 3 min and 50% ethanol solution as the extraction solvent). This is also confirmed by the statistical analysis of the results, where a statistically negative influence of ethanol content on the TPC yield was observed in the US-treated samples with a confidence interval of 95% (*p* < 0.05). A trend of decreasing TPC yields with increasing ethanol content was also observed in blackthorn fruit extracts [24], where TPC yields were 4.64–6.85 times lower when using a 100% ethanol solution as the extraction solvent than when using lower concentrations of ethanol solutions. In general, the TPC yields of US-treated samples are lower than the TPC yields obtained in the study by Ebrahimi et al. (2024) with the same plant material and using the same extraction method [25]. In the mentioned study, depending on the solvent used (0, 25, 50, 75 and 100% ethanol solution), the treatment time (4, 6 and 8 min), the amplitude used (25, 30 and 35%) and the ratio of plant material and solvent (1, 2, 3, 4 and 5% *w*/*v*), TPC yields between 2.05 ± 0.10 and 12.12 ± 0.09 mmol GAE/L (approx. 38.78 to 45.80 mg/g_d.m._) were obtained. Considering the ratio between the diameter of the US probe; the volume of the extraction mixture, which is almost twice as large in the aforementioned work (2 mm for 10 mL vs.12 mm for 100 mL); and the shape of the laboratory dishes (Falcon Tubes of 15 mL vs. laboratory beakers of 250 mL), the yields obtained are lower, as expected. The diameter of the probe is one of the most important parameters that directly influences the efficiency of US extraction. In general, when using smaller diameter probes, the effect of transient acoustic cavitation is greater, but within a narrow range. In contrast, probes with a larger diameter distribute the energy over a larger area, so that the effect of the US is stronger.

Compared to the results of the US extraction, the yield of TPC after the HVED-assisted extraction was up to 17.34 times lower (Figure 1b). In particular, the highest yield of TPC in HVED-treated samples was observed in sample 0LHA3 (with deionized water as the extraction solvent, argon as the applied gas, an applied voltage of 20 kV and a treatment time of 9 min) and was 7.04 ± 0.00 mg GAE/g_d.m._, while the lowest amount was 0.95 ± 0.14 mg GAE/g_d.m._ for the 50LHN5 sample (with 50% ethanol solution as the extraction solvent, nitrogen as the applied gas, an applied voltage of 20 kV and a treatment time of 3 min). Although the highest and lowest yields were obtained using different gases, no statistically significant influence of the gases was found (*p* > 0.05). In contrast, treatment time, applied voltage and ethanol content showed a statistically significant impact on the TPC yield in the HVED-treated samples. In particular, a longer treatment time had a positive effect on the TPC yield, while the ethanol content and the applied voltage had a negative effect, which consequently led to a lower TPC yield. Comparing different treatment times (3, 6 and 9 min) at the same applied voltage (e.g., 20 kV), gas (e.g., argon) and ethanol content (e.g., deionized water as the extraction solvent), the positive effect of the treatment time was clearly visible. Specifically, a TPC yield of 2.01 ± 0.03 mg GAE/g_d.m._ was observed under the given conditions with a treatment duration of 3 min. A longer HVED treatment increased the TPC yield to 4.68 ± 0.00 mg GAE/g_d.m._ for a 6 min treatment, and 7.04 ± 0.00 mg GAE/g_d.m._ for a 9 min treatment. On the other hand, a higher ethanol content and an increase in the applied voltage from 20 kV to 25 kV had a negative effect on the TPC yield, regardless of the gas used. Some phenolic compounds are bound to the cell membrane, and it is necessary to invest a little more energy to extract them. For this reason, a longer treatment time has a positive effect on the yield, as a longer treatment time invests more energy and thus increases the chance that the bonds between the phenolic compound and the cell wall are broken and the phenolic compounds become more available for extraction. In contrast, a higher applied voltage causes the formation of a greater amount of molecular oxygen, which ultimately leads to the formation of molecular ozone, which negatively affects the phenolic compounds by breaking the bonds in their aromatic rings. The negative effect of increasing the voltage was also observed in samples of green tea powder, where a decrease in the TPC yield of 2.68–7.66 mg GAE/L was observed at the same treatment time but at a higher voltage (25 kV vs. 20 kV) [26].

In parallel to the non-thermal extraction methods, a conventional thermal extraction method at 60 °C was also performed and the obtained TPC yields are shown graphically (Figure 1c). In general, the TPC yields obtained after thermal extraction with deionized water as the extraction solvent were significantly lower than the yields obtained with the same solvent but with non-thermal extraction methods (US and HVED). However, when ethanol solutions were used, higher yields were observed compared to the HVED-treated samples, but lower compared to US-treated samples. Ethanol content was a statistically significant variable of thermal extraction with a positive influence on the TPC yield (*p* < 0.05). This trend is opposite to the trends observed for US- and HVED-treated samples. The highest TPC yield of 9.58 ± 0.28 mg GAE/g_d.m._ was observed in the 50LT6 thermally treated sample (with 50% ethanol solution as the extraction solvent and a treatment time of 6 min), while the lowest yield of 0.16 ± 0.01 mg GAE/g_d.m._ was observed in the 0LT9 sample (with deionized water as the extraction solvent and a treatment time of 9 min). Considering only the 0LT thermally treated samples (with deionized water as the extraction solvent), it was found that the TPC yield decreased with longer treatments (up to 6.25 times). This is consistent with the heat sensitivity of phenolic compounds, which decompose or degrade at high temperatures and longer treatment times, leading to a decrease in TPC yield [27]. A decrease of 89% in the high-temperature treatment at 110 °C (compared to the treatment at 70 °C) was also observed in samples of myrtle leaves [28].

### 2.2. Vitexin Content

Regardless of the applied extraction method, the yield of vitexin decreases statistically significantly with increasing ethanol content (*p* < 0.05). Accordingly, the lowest vitexin yield was observed when a 50% ethanol solution was used as the extraction solvent. Specifically, a decrease in the vitexin yield of 2.77–5.02 times was observed in the US-treated samples, of 2.12–6.72 times for the HVED-treated samples and of 3.38–4.87 times for thermally treated samples. By optimizing the process parameters of the mentioned extractions and to achieve an optimal yield of vitexin, the ethanol content in the extraction solvent should be 0%. The obtained results correlate with the very poor solubility of the mentioned compound in ethanol and similar solvents. Due to its chemical structure (apigenin-8-C-glucoside), the flavonoid vitexin is highly soluble in organic solvents such as dimethyl sulfoxide (DMSO), but significantly less soluble in water and ethanol [29]. The results of vitexin yield for each extraction method are shown graphically (Figure 2).

The highest yield of vitexin in the thermally treated samples was 1369.91 ± 19.82 ng/mL, and was recorded in the 0TL3 sample (with deionized water as the extraction solvent and a treatment time of 3 min). Temperatures between 50 and 70 °C were found to be optimal temperatures for the extraction of vitexin from dried Mas Cotek leaves [30]. Namely, gentle heating has the ability to soften the plant tissue, weaken the integrity of the cell wall and consequently promote the release of bound active compounds from the plant material [27]. In order to achieve optimal yield, the optimal input parameters of thermal extraction were determined by statistical projection and corresponded to the parameters of the 0TL3 sample, which also showed the highest yield of vitexin. These parameters for thermal extraction are also the most acceptable from an energetic and environmental point of view, as the extraction takes the least time (and therefore the carbon footprint is the lowest) and the solvent used is also environmentally friendly.

Regarding the non-thermal extraction methods, the yield of vitexin was slightly higher in the HVED-treated samples than in the thermally treated samples. In particular, the highest yield was observed in the 0LHN3 sample (with deionized water as the extraction solvent, using nitrogen gas, an applied voltage of 20 kV and a treatment time of 9 min) and was 1481.22 ± 5.70 ng/mL. In addition to the extraction solvent, the applied voltage also had a statistically significant influence on the yield of vitexin in the HVED-treated samples (*p* < 0.05), while the application of different gases (argon/nitrogen) had no statistically significant influence on the yield of vitexin (*p* > 0.05). As well as a higher ethanol content, the application of higher voltages (25 kV) in the HVED-treated samples also had a statistically negative effect on the vitexin yield. To achieve the optimum values, the treatment time was statistically determined at 6.25 min, the ethanol content at 0%, the applied voltage at 20 kV and nitrogen gas was used. A decrease in the yield of flavonoids (by 558.54–2552.22 µg quercetin/mL) due to the application of a higher voltage (from 20 to 25 kV) was also observed in green tea powder samples [26]. In addition to TPC, higher voltage also has a negative effect on the yield of flavonoid compounds such as vitexin. The increased formation of molecular ozone and the breaking of bonds in their aromatic rings leads to a decrease in the yield of the flavonoid compound. Furthermore, exposure to higher voltages potentially leads to denaturation of the enzyme phenylalanine ammonialyse, an enzyme essential for the synthesis of flavonoid compounds [31].

In contrast to the HVED-treated samples, the US-treated samples showed the lowest yield of vitexin. Although the obtained values are the lowest, they were not significantly different from the vitexin yield results obtained with other extraction methods, which ranged from 193.15 ± 5.60 ng/mL to 969.97 ± 3.52 ng/mL. The achieved yields are well below the yields determined in the study by Maravić et al., 2022 [23]. However, taking into account the ratio of plant material to solvent in the aforementioned study (1:10 compared to 1:50 in this work) as well as the treatment time of 30 min (and up to ten times longer) and the temperature of 50 °C (and up to 20 °C higher), such differences in yields can be explained [23]. Consequently, a longer treatment time with a higher concentration of plant material and the application of a higher temperature leads to a stronger formation of broken bonds between flavonoids and sugars, making the vitexin more available for the solvent. In this work, the yield of vitexin is statistically influenced by the input variables amplitude and the mutual interaction of amplitude and ethanol content (*p* < 0.05). In contrast to the statistically negative influence of the ethanol content and the mutual negative interaction between the applied amplitude and the ethanol content, the input variable amplitude has a positive effect on the vitexin yield. Specifically, by applying the same solvent for the same treatment time, increasing the amplitude increases the yield of the desired component, i.e., vitexin. The trend of a statistically significant increase in vitexin yield with increasing amplitude was also observed in samples of *Trollius chinensis* flowers [32]. As the US amplitude increases, the energy of the US wave increases, making the phenomenon of transient acoustic cavitation more prominent. Bubbles in the liquid medium implode and subsequently lead to large shear forces in the liquid medium, turbulence and mixing of the sample. Among other things, this leads to sonoporation and fragmentation of the plant material, making the desired components (e.g., vitexin) more available for the solvent and thus for extraction [33,34,35].

### 2.3. Determination of Antioxidant Capacity (AC)

#### 2.3.1. Ferric Reducing Antioxidant Power (FRAP) Assay

One of the methods for measuring AC is the FRAP method, a very fast, cheap and simple spectrophotometric method that does not require any special equipment. However, the results obtained do not represent “real” values, as antioxidants, such as some proteins and carotenoids, which act on the principle of hydrogen transfer, cannot be determined [36,37]. In addition, antioxidants that contain a thiol group (-SH) in their structure, such as glutathione, cannot be detected using this method either [38]. Since the reaction itself requires a redox potential lower than the Fe^3+^/Fe^2+^ redox potential, many compounds (not necessarily antioxidants) react with a lower redox potential, resulting in false high positive values of AC [39]. Regardless of the redox potential, some compounds still absorb at the determination wavelength due to their chemical composition, which at the same time has no antioxidant properties, which also leads to false positive values of AC [40]. On the other hand, some compounds such as quercetin, tannic acid, caffeic acid and ferulic acid react more slowly, so they are not able to reduce Fe^3+^ due to the shorter reaction time, so the results obtained are lower than the actual AC values [41]. Despite these shortcomings, the AC of the samples was measured and the results obtained are presented graphically (Figure 3).

The highest AC values were found in the US-treated samples, where a range of 1677.16 ± 36.98 to 2290.50 ± 56.69 µM Fe^2+^ was observed depending on the applied amplitude, extraction solvent and treatment time. Although no input parameter had a statistically significant effect (*p* > 0.05) on the CA value of the US-treated samples, the highest values were observed when a 25% ethanol solution (extraction solvent) was used. The highest value was recorded for the 25LU3 sample, where the AC value of 2290.50 ± 56.69 µM Fe^2+^ was obtained with 25% ethanol solution as the extraction solvent at an amplitude of 50% over a treatment time of 6 min. The application of higher amplitudes may lead to the formation of smaller air bubbles which directly affect the extraction efficiency and reduce the efficiency due to the lower transfer of ultrasonic energy to the solvent [42]. Although it did not statistically significantly affect the AC value of the US-treated samples, treatment time is an important extraction factor. A longer extraction time can lead to a higher yield of compounds with AC, but too long treatment has a negative effect on it by degrading it [43].

Compared to the US-treated samples, lower AC values were found in the thermally treated samples (845.71 ± 16.42 to 2063.55 ± 34.48 µM Fe^2+^). In contrast to the US-treated samples, the input variable, ethanol content, had a statistically significant influence (*p* < 0.05) on the AC value in the thermally treated samples. A negative influence of treatment time on the ACs of the thermally treated samples when deionized water was used as the extraction solvent was also observed. Specifically, by increasing the treatment time from 3 to 9 min, the AC value decreased 1.50 times. In contrast, the AC value of the thermally treated samples increased when ethanol solutions were used and the treatment time was increased. The use of higher ethanol concentrations does not necessarily guarantee higher AC values. Safdar et al., 2017, reported higher AC values of kinnow peel samples treated with 80% alcohol solutions than with 100% alcohol solutions [44].

The lowest AC values were found in the HVED-treated samples, which ranged from 912.75 ± 28.75 to 1898.20 ± 4.03 µM Fe^2+^. A statistically significant influence on the AC value of the HVED-treated samples was found for all input variables except the variable of applied gas. Although 61.11% of the LHA samples had a higher AC value, this variable did not have a statistically significant effect on the AC value (*p* > 0.05). The treatment time had a statistically positive influence (*p* < 0.05) on the AC value, while the ethanol content and the applied voltage had a negative influence on the same value (*p* < 0.05). The obtained results do not correlate with the results of studies on HVED-treated samples of thyme leaves [45] and olive leaves [46]. In particular, in the study using thyme leaves, the highest AC value was found when argon gas was applied, and a positive influence of the applied voltage and ethanol content was observed as statistically significant input variables (*p* < 0.05) [45]. Considering the different sample types and their chemical composition, the obtained results are not unexpected (the TPC yield of thyme leaf samples is 6.04 times higher than the TPC yield of sugar beet leaf samples). Namely, the sugar beet leaves are a good source of proteins and bioactive compounds with antioxidant activity that work on the principle of hydrogen transfer and therefore cannot be determined by this method and consequently cannot contribute to the AC results [36,37,47].

#### 2.3.2. DPPH (2,2-Diphenyl-2-picrylhydrazyl) Free Radical Assay

The DPPH method is another spectrophotometric method of determining the antioxidant capacity of samples. Similar to the FRAP method, it is a quick, simple and inexpensive method that does not require any special equipment. Like any in vitro method for the determination of AC, this method is subject to certain limitations. Since the solubility of DPPH• radicals is limited to organic solvents (ethanol/methanol), the results of measuring the ACs of hydrophilic compounds with antioxidant activity are questionable [48]. Furthermore, as with the FRAP method, certain compounds absorb at the same measurement wavelength, which contributes to high false positive AC values. Due to the spatial limitation of the reaction between DPPH• radicals and large molecules with antioxidant activity, the reaction proceeds slowly or not at all, resulting in falsely lower AC values [49]. In addition, it is not suitable for the determination of AC in samples with proteins (which also possess antioxidant activity), as protein precipitation occurs in alcoholic solutions, which also leads to falsely low AC values [50]. Nevertheless, the AC value was determined in all of the samples using the DPPH method and the obtained results are presented graphically (Figure 4).

Regardless of the extraction method used, the AC values determined by the DPPH method were significantly lower than those measured by the FRAP method. In particular, values that were 2.69–5.15 times lower were measured in the US-treated samples, 3.06–7.95 in the thermally treated samples and 2.95–13.40 in the HVED-treated samples. These observations are consistent with studies on US-treated samples of makiang seeds, where the AC values measured by the DPPH method were up to 1.51 times higher than those measured by the FRAP method [51]. In contrast, HVED-treated thyme samples showed higher values (by more than 15 times) of the AC measured by the DPPH method [45]. As already mentioned, the chemical composition of the plant material has a great influence on the final value of the AC and thus on the relationship between the determination methods used. In this work, although the AC values measured by the DPPH method were lower, the trend of increase/decrease between extraction methods and within samples for each extraction method was similar to the trend determined by the FRAP method. In particular, no single statistically significant influence of the input variables was found in the US-treated samples, but a statistically negative influence of the mutual interaction of the amplitude and the ethanol content on the AC value was determined by the DPPH method (*p* < 0.05). The highest AC value (in general, but also within the US-treated samples) of 757.29 ± 1.82 µM TE was observed in the US-treated 25LU9 sample (with 25% ethanol solution as the extraction solvent, an applied amplitude of 50% and a treatment time of 9 min). An increase in the AC value using 25% ethanol solution (extraction solvent) was also observed. The AC values of the US-treated samples ranged from 292.07 ± 7.32 to 757.29 ± 1.82 µM TE and increased 1.50 times compared to the HVED-treated samples and 1.16 times compared to thermally treated samples. The obtained results are not consistent with the research results on olive kernel samples. In fact, in the aforementioned study, higher values of the AC were measured with the HVED extraction method (at a voltage of 40 kV) than with pulsed electric field or ultrasound methods [52]. The application of a significantly higher voltage (15 or 20 kV higher) contributed to the appearance of more expressive electrical charges, which consequently led to a stronger and more expressive fragmentation of the plant material. This had a positive effect on the destruction of complexes between phenolic compounds and other compounds (such as sugars, proteins, etc.) and contributed to increasing the yield of components with antioxidant activity [53].

In the samples of sugar beet leaves treated with HVED, the treatment time proved to be a statistically significant positive input variable for the AC value measured by the DPPH method (*p* < 0.05). Similarly to the FRAP method, the gas used had no statistically significant effect on the value of the AC (*p* > 0.05), with a higher AC value found in 72.22% of the samples in which argon gas was used. The obtained results correlate with the results observed in HVED-treated thyme leaf samples [45]. In addition to the statistically positive influence of the treatment time, the interaction between the treatment time and the gas used had a statistically significant negative influence on the AC value (*p* < 0.05). In general, the reasons for the lower AC value in the HVED-treated samples (compared to the US-treated samples) have already been explained, and the values ranged from 133.00 ± 10.30 to 504.14 ± 1.82 µM TE.

A similar range of values (109.21 ± 4.55 to 655.57 ± 103.06 µM TE) was found for the thermally treated samples. The highest AC values of 655.57 ± 103.06 µM TE were recorded in the TL25/9 sample (with 25% ethanol solution as the extraction solvent and a treatment time of 9 min) and the lowest values of 109.21 ± 4.55 µM TE were recorded in the TL0/6 sample (with deionized water as the extraction solvent and a treatment time of 6 min). The treatment time showed no statistically significant influence on the AC value (*p* > 0.05), but the ethanol content did (*p* < 0.05). As with the determination of AC by the FRAP method, the highest values of AC determined by the DPPH target in thermally treated samples were observed when a 25% ethanol solution was used as the extraction solvent. The recorded results are consistent with the research results on thermally treated (100 °C) samples of white mulberry. Namely, with a longer treatment time (from 15 to 45 min) and the application of a higher ethanol content (from 0% to 70% ethanol solution), the AC value measured by the DPPH method, expressed as radical scavenging (%), increased by 1.24 to 1.59 times [54].

#### 2.3.3. Determination of Antioxidant Capacity by Electron Spin Resonance (ESR) Spectroscopy

Figure 5a shows the results of the ESR measurements obtained in US-treated samples (from 118.42 ± 7.17 to 280.93 ± 7.32 µM TE), using various amplitudes, treatment times and ethanol contents. It can be seen that the US-treated samples with deionized water as the extraction solvent (without ethanol) had lower AC values compared to the other US-treated samples, with the exception of the 0LU6 (227.26 ± 5.15 µM TE), 0LU8 (201.53 ± 6.18 µM TE) and 0LU9 (174.71 ± 16.79 µM TE) samples. These samples were US-treated at maximum amplitude (100%) for 9, 6 and 3 min, respectively. In general, all of the input variables showed a positive statistical significance on the value of the antioxidant capacity, while on the other hand, the interaction between the applied amplitude and ethanol content had a statistically significant negative effect on the AC values (*p* < 0.05). In a previous work [55], the influence of the sonication treatment of water on the formation of hydroxyl radicals was investigated by the EPR spectroscopy/spin-trapping method. It was found that the number of radicals increased with the treatment time and with an increasing amplitude. The results in Figure 5a confirm previous findings. Based on these facts, it can be assumed that the mentioned samples contained a larger number of hydroxyl radicals compared to other US-treated samples with deionized water. Although hydroxyl radicals are short-lived, it is possible that they reacted with some of the components extracted from the sample, resulting in the formation of stable radicals. Such stable radicals could contribute to the increase in the AC value. US-treated samples with an ethanol solution (extraction solvent) showed slightly higher AC values than samples extracted with deionized water. There are two possible explanations for this result: first, the use of a water–ethanol mixture may lead to better extraction and consequently to a higher concentration of antioxidant compounds; and second, the addition of small amounts of ethanol to water leads to the appearance of 1-hydroxyethyl free radicals [56]. These radicals could react with DPPH radical and contribute to an increase in the AC value in the US-treated samples. One could expect that the concentration of free radicals formed from ethanol will increase by increasing the ethanol content in the extraction solvent. However, it has been shown that once the optimal ethanol content is reached, a further increase leads to a decrease in the number of carbon radicals [57]. Therefore, a higher ethanol content, which is particularly pronounced in sample 50LU8 (with a 50% ethanol mixture, 100% amplitude and 9 min treatment time), could lead to a lower number of free 1-hydroxyethyl radicals, as in the case of carbon radicals [57].

In the earlier work mentioned above [55], the influence of HVED-treated water on the formation of hydroxyl radicals was also investigated. It was proven that the plasma-induced formation of hydroxyl radicals depends on several factors, and one of the most important is the type of gas used. Therefore, two series of samples were investigated in the presence of nitrogen (from 53.23 ± 2.58 to 199.21 ± 6.55 µM TE) and argon (from 34.25 ± 2.76 to 181.78 ± 3.57 µM TE) in this work. The results are shown in Figure 5b. Statistically, a significant positive effect (*p* < 0.05) of the treatment time on the AC value was observed for all of the HVED-treated samples, regardless of the gas used. Argon reacts with plasma electrons and then reacts with water molecules to form hydroxyl radicals [58]. Nitrogen molecules react in a very similar way [59]. The highest concentration of hydroxyl radicals was found when using argon mixed with water as a working gas, followed by nitrogen [58]. As already indicated, the addition of even small amounts of ethanol to water leads to the occurrence of carbon radicals, and the concentration increased sharply with further increases in ethanol content, indicating that ethanol molecules are predominantly degraded to carbon radicals [57]. These radicals were presumably primary radicals that could react with other radicals, including DPPH, and contribute to an increase in the AC value of the HVED-treated samples. The results also showed a continuous increase in radical concentration with the increase in ethanol content until the value χ = 0.37 was reached, after which the value started to decrease. All of these facts should be considered when analyzing the results. The 0LHN3 sample, which was HVED-treated with deionized water as an extraction solvent for 9 min at a lower voltage (20 kV) in the presence of nitrogen gas, showed the highest AC value (199.21 ± 6.55 µM TE). This is followed by the 0LHA3 sample (181.78 ± 3.57 µM TE) treated under the same conditions in the presence of argon. However, these two samples were an exception compared to other HVED-treated samples with deionized water (extraction solvent). The addition of ethanol in a lower amount (25%) indicates a positive effect when coupled with argon and a higher voltage (25 kV), while this effect was absent under the same conditions with nitrogen. Under these conditions, argon obviously plays the most important role. In contrast, the presence of nitrogen seems to favor a higher ethanol content with the longest treatment time regardless of the applied voltage (except for Sample 50LHA5). From the presented results, each factor can have a different influence on the AC under different conditions. To determine which factor predominates, further measurements would need to be performed.

The effects of thermal treatment on the ESR results are shown in Figure 5c. The temperature influence, regardless of the treatment time (from 3 to 9 min), did not show a noticeable effect on the AC value of the thermally treated samples when deionized water was used as an extraction solvent. However, when all of the thermally treated samples were considered, the treatment time, ethanol content and the interaction between treatment time and the ethanol content showed a positive statistical significance on the AC value (*p* < 0.05). With the addition of the 25% ethanol solution, a slight increase in the AC value can be observed with an increasing treatment time. This effect was more pronounced with an ethanol content of 50%. The maximum AC value (331.36 ± 0.32 µM TE) was reached at the longest treatment time (9 min) with 50% ethanol solution (extraction solvent). This result indicates that ethanol in combination with deionized water improves the extraction of the bioactive components. The positive influence of thermal treatment could be attributed to the increase in the TPC and total flavonoid content (TFC), as shown in the garlic samples [60]. In contrast to the results of that study, this study’s results indicate a positive correlation between treatment time and AC (*p* < 0.05). However, it should be noted that the treatment times in this study were shorter, namely 3–9 min, compared to 15–60 min.

### 2.4. Plasma Characterization by Optical Emission Spectroscopy (OES)

Optical emission spectra (OES) are shown in Figure 6 of (a) argon and (b) nitrogen plasma. In each case, the spectra of sugar beet leaves immersed in water (black line), 25% ethanol (red line) and 50% ethanol (blue line) during plasma treatment are shown. All of the emitted lines were identified using the NIST Atomic Spectra Database [61] or consulting the literature [62,63,64,65]. In Figure 6a, the Ar plasma is shown with the dominant emission from neutral atomic argon and oxygen (the emission lines are saturated), while atomic hydrogen from the Balmer series is present with a distinctive emission line at 656 nm, ascribed to Hα. A weak molecular hydrogen emission (Fulcher band) is observed around 600 nm. Qualitatively, there are no significant differences between the spectra for water, 25% ethanol and 50% ethanol. The presence of atomic oxygen and hydrogen in all of the spectra indicates that plasma interacts mainly with the air molecules dissociating hydrogen and oxygen molecules and with the argon atoms emerging from the liquid. Figure 6b shows nitrogen plasma in which the emission of nitrogen species dominates, while atomic hydrogen is also present. It is evident that in the blue part of the spectrum emission, molecular N_2_ (2nd positive band) and N_2_^+^ (1st negative band) are present, while across the entire optical spectrum, atomic and ionic nitrogen is observed. Atomic hydrogen shows a clear emission at 656 nm. Again, no differences are observed between the spectra recorded for deionized water, 25% ethanol and 50% ethanol. In both cases, in argon and nitrogen plasmas, the emission of carbon, as a possible product of ethanol dissociation, was not observed. This indicates that the plasma does not interact with the ethanol in manner that decomposes the structure of the liquid.

## 3. Materials and Methods

### 3.1. Chemicals

All of the chemicals used in this work corresponded to p.a. purity grade or higher. For HPLC-MS/MS analysis, deionized water was purified using a water purification system from Siemens Ultra Clean (Munich, Germany); otherwise, deionized water from the Faculty of Food Technology and Biotechnology, Zagreb, Croatia, was used. Anhydrous sodium carbonate was purchased from K.T.T.T. (Sveta Nedjelja, Croatia). Gallic acid, 6-hydroxy-2,5,7,8-tetramethylchroman-2-carboxylic acid (Trolox) and 2,4,6-Tripyridyl-S-triazine (TPTZ) standards were purchased from Acros Organics (Geel, Belgium). Vitexin standard (4′,5,7-trihydroxyflavone-8-glucoside) and formic acid for mobile phase were purchased from Sigma-Aldrich (St. Louis, MO, USA). Methanol, acetonitrile and iron (II)-sulfate heptahydrate were bought from Honeywell (Seelze, Germany). Iron (III) chloride, sodium acetate trihydrate and ethanol were purchased from GRAM MOL (Zagreb, Croatia). 2,2-Diphenyl-2-Picrylhydrazyl (DPPH) was bought from ABCR (Karlsruhe, Germany) and stored in the dark at 4 °C. Folin-Ciocalteu reagent purchased from Kemika (Zagreb, Croatia) was stored under the same conditions. Hydrochloric acid was bought from Carlos Erba Reagents (Val-de-Reuil, France) and glacial acetic acid was purchased from J.T. Baker (Gliwice, Poland).

### 3.2. Plant Material

Dry sugar beet (*Beta vulgaris* L.) leaves with a certain dry matter of 94.49 ± 1.6% were provided by project partners from Turkey (Kayseri Şeker, Kocasinan Kayseri, Turkey), the same as in the previous study [47], and in order to facilitate extraction during sample preparation, dry leaves were ground to the particle size distribution of d(0.1) ≤ 238.490 µm; d(0.5) ≤ 630.116 µm; d(0.9) ≤ 1196.769 µm. The particle size was determined using the laser particle size analyzer Mastersizer 2000 (Malvern Instruments GmbH, Herrenberg, Germany).

### 3.3. Extraction Methods and Sample Labeling

#### 3.3.1. High-Power Ultrasound Assisted Extraction (US)

A total of 27 samples of sugar beet leaves were ultrasonically treated with Q700 Sonicator (Qsonica, Newtown, CT, USA) by adding 100 mL of extraction solvent (deionized water and 25% and 50% ethanol solutions) into a 250 mL double-walled laboratory beaker with 2 ± 0.001 g of weighed ground sugar beet leaves. The position of the ultrasonic probe (d 12 mm) is described in detail by Dukić et al. [19]. To avoid overheating, high-capacity recirculating chiller 4905 (Qsonica, Newtown, CT, USA) was used. The name of the sample consists of an alphanumeric designation, whereby the number before the letter indicates the value of the ethanol content (0, 25 and 50%), the letters indicate the type of treatment (LU, sugar beet leaves US- treated) and the number after the letters indicates the specified US process parameters. Sample 25LU4, for example, represents a sample that was US-treated with 25% ethanol as an extraction solvent and with an amplitude of 50% over a treatment time of 9 min. For a better understanding, the labeling of the samples and the US process parameters are listed in Table 1.

#### 3.3.2. High-Voltage Electrical Discharge-Assisted Extraction (HVED)

In addition to the US method, another non-thermal extraction method was used. To compare non-thermal and thermal methods, high-voltage electrical discharge-assisted extractions (HVED) were performed as well. A total of 1 ± 0.001 g of ground, dry sugar beet leaf samples was mixed with 50 mL of extraction solvent (deionized water and 25% and 50% ethanol solutions) in a 100 mL reactor. A total of 36 samples of sugar beet leaves were HVED-treated using the IMP-SSPG-1200 generator (Impel group d.o.o., Zagreb, Croatia). A frequency of 100 Hz, a pulse width of 400 ns, a distance between the electrodes of 15 mm and a gas flow of 0.75 L/min were previously described in detail by Nutrizio et al. [66]. The applied HVED treatment parameters were a voltage of 20 and 25 kV for nitrogen and argon gas over a treatment time of 3, 6 and 9 min. Same as with the US-treated samples, the sample name consists of an alphanumeric designation, where the number before the letter indicates the ethanol content value (0, 25 and 50%), the letters LH indicate the HVED treatment, the letter A (stands for “argon”) or N (stands for “nitrogen”) indicate the used gas and the number after the letters indicates the HVED process parameters. For example, Sample 0LHN1 represents an HVED-treated sample with deionized water as extraction solvent and nitrogen as applied gas over a treatment time of 6 min at 25 kV. For a better understanding, the names of the samples and the HVED process parameters are listed in Table 2.

#### 3.3.3. Conventional Thermal Extraction

The thermal extraction of bioactive compounds and antioxidant capacity from sugar beet leaves using the conventional method was performed using a water bath with a mixing function (Memmert GmbH, Schwabach, Germany). The temperature in the water bath was set to 60 °C and stirring was carried out at 150 rpm. A total of 2 ± 0.1000 g of ground dry sugar beet leaves was weighed into a 250 mL laboratory beaker and 100 mL of extraction solvent (deionized water and 25% and 50% ethanol solutions) was added. The samples were placed in the water bath and left in the water bath for 3, 6 and 9 min. The labeling of the thermally treated samples is shown in Table 3.

### 3.4. Analysis

#### 3.4.1. Determination of Total Phenolic Content (TPC)

The total concentration of phenolic compounds in the samples was measured with the UV-VIS spectrophotometer UV-2600i (Shimadzu, Kyoto, Japan) at a wavelength of 765 nm and according to the Folin–Ciocalteu method [67], using gallic acid (GAE) as a standard and calculated using the following linear equation:y = 0.00107826x + 0.111618(1)
where “y” represents the measured absorbance (at 765 nm), and “x” the equivalent GAE concentration (in mg/L). The sample preparation and measurement procedure are described in detail in a previous article [19]. Taking into account the known mass of the sugar beet leaves and the percentage of dry matter (94.49 ± 1.6%), the results were expressed in mg/g_d.m_.

#### 3.4.2. Determination of Vitexin Content

##### Standard Solutions and Samples for HPLC-MS/MS Determination

The corresponding amounts of the reference substances were dissolved in methanol to obtain the stock solutions for vitexin. The working solutions used for further investigations were prepared by diluting the stock solutions with methanol or the mobile phase. All of the solutions and samples prepared for high-performance liquid chromatography coupled with mass spectrometry (HPLC-MS/MS) analysis were filtered through a 0.20 μm filter syringe (Filtres Fioroni, Senigallia, Italy) before use. All of the samples were injected in duplicate.

##### Chromatographic and Mass Spectrometric Conditions

The chromatographic analysis was carried out using an Eksion LC HPLC system (SCIEX, Framingham, MA, USA). The analytes were separated on a Phenomenex column Luna Omega 3 μm Polar C18 100 Å, 100 × 4.6 mm (Torrance, CA, USA), with thermostat column temperature of 40 °C, automated sampling temperature of 4 °C and injection volume of 10 µL. The mobile phases consisted of: A 100% H_2_O with 0.01% HCOOH (*v*/*v*) and B 100% ACN with 0.01% HCOOH (*v*/*v*) at a flow rate of 0.40 mL/min. The gradient was set as follows: 1 min 10% B, 13 min 90% B, 15 min 90% B, 16 min 10% B. Column efflux was monitored using a SCIEX 4500 QTRAP (SCIEX, Framingham, MA, USA) triple quadrupole mass spectrometer with an electrospray ionization (ESI) source. Vitexin was ionized in negative electrospray ionization mode, with ionization temperature of 300 °C, ion spray voltage of −4500 V, drying gas temperature of 190 °C and drying gas flow of 9.0 L/min [68]. Nitrogen was used as nebulizing, curtain and collision gas. The mass spectrometer was programmed for multiple reaction monitoring (MRM). The precursor ion [M − H]—at *m*/*z* 431.0 was monitored via the first quadrupole filter (Q1), while the product ion at *m*/*z* 311.1, 283.1 and 117.0 was monitored via the third quadrupole filter (Q3) [69]. Declustering potential (DP) was −120 V, the collision energy (CE) was −32.0 and the collision cell exit potential (CXP) was −11.0. The peak areas obtained from MRM were used for the quantification of vitexin [70]. The data were processed with Multiquant 3.6 software (SCIEX, Darmstadt, Germany).

##### Method Validation

For the chromatographic determination’s linearity test, standard vitexin solutions were injected in triplicate in seven steps over a concentration range of 0.10 to 100 ng/mL. Good linearity was obtained, with correlation coefficients between 0.99956 and 0.9987. The limit of detection (LOD) and the limit of quantification (LOQ) were investigated by stepwise dilution of each standard solution with methanol; LOD and LOQ were calculated for a ratio S/N of 3 and 10, respectively. The LOQ for chromatographic determination of vitexin quantification was confirmed by injecting the 6-fold vitexin standard with a concentration of 0.50 ng/mL, i.e., the second point of the calibration curve. The recovery was between 90.34% and 110.63%; the average recovery was 105.70%. The standard deviation (SD) was 0.0377 ng/mL. The repeatability and reproducibility of the method was expressed as relative standard deviation (RSD) and amounted to 7.44%.

The determination of vitexin content in the samples was calculated using the following linear equation:y = 28480.20214x(2)
where “y” represents the area under the peak and “x” the equivalent vitexin concentration (in ng/mL).

#### 3.4.3. Determination of Antioxidant Capacity (AC)

##### Ferric Reducing Antioxidant Power (FRAP) Assay

This method is based on the reduction reaction of the yellow-colored complex iron-2,4,6-tripyridyl-s-triazine (TPTZ) in the presence of antioxidants, resulting in a blue-colored product. The reaction takes place in an acidic medium at pH = 3.6 to ensure good solubility of the iron. At lower pH values, the ionization potential, which enables electron transfer, decreases, and at the same time the redox potential increases, which additionally promotes the reaction in the direction of electron transfer. The redox potential of the Fe(III)/Fe(II) reaction is 0.77 V, and all compounds with a lower redox potential participate in the iron reduction reaction and thus contribute to the final result of AC [71].

The AC in the samples was measured with the UV-VIS spectrophotometer UV-2600i (Shimadzu, Kyoto, Japan) at a wavelength of 595 nm and according to the FRAP method [46] using FeSO_4_·7H_2_O as a standard and calculated using the following linear equation:y = 6.81142∗10^−4^x + 0.00843793(3)
where “y” represents the measured absorbance (at 595 nm) and “x” the equivalent Fe^2+^ concentration (in µM). The chemicals/sample preparation and measurement procedure are described in detail in a previous article [46].

##### DPPH (2,2-Diphenyl-2-picrylhydrazyl) Free Radical Assay

This spectrophotometric method for the determination of AC is based on the reduction of the 2,2-diphenyl-2-picrylhydrazyl (DPPH) radical in a methanol solution. The DPPH radical shows strong absorption in the visible region of the spectrum (515 nm) due to the unpaired electron. In the presence of an electron donor, an antioxidant that “quenches” free radicals, DPPH radicals are stabilized by electron pairing, resulting in a color change of the solution from purple (oxidized form of the radical) to yellow (reduced form of the radical). AC is measured by the change in absorbance of the reaction mixture at a given time compared to the same effect obtained with a known reducing agent [72].

The AC in the samples was measured with the UV-VIS spectrophotometer UV-2600i (Shimadzu, Kyoto, Japan) at a wavelength of 515 nm and according to the modified DPPH method [73] using Trolox as a standard and calculated using the following linear equation:y = 0.0007x + 0.0032(4)
where “y” represents the measured absorbance (at 515 nm) and “x” the equivalent Trolox (TE) concentration (in µM). To determine the AC, 100 μL of the sample was pipetted into a test tube and 3.9 mL of a 0.094 mM DPPH solution was added and shaken. The reaction took place in the dark for 30 min (time to reach equilibrium). The measured absorbance was compared with the absorbance of the blank sample (which contained methanol instead of the sample).

##### Determination of Antioxidant Capacity by Electron Spin Resonance (ESR) Spectroscopy

ESR spectroscopy is a technique that enables the direct detection of free radicals in the sample and is therefore very suitable for measuring the antioxidant capacity of different types of samples [74,75]. The ESR measurements have been performed at Ruđer Bošković Institute in Zagreb, Croatia, at room temperature on a Bruker EMX CW spectrometer (Bruker, Billerica, MA, USA) equipped with the Varian V-7300 magnet (Palo Alto, CA, USA) and Bruker microwave bridge ER 041 XG. The spectroscopic parameters were microwave frequency of 9.3 GHz, power attenuation of 10 mW, magnetic field modulation amplitude of 1 G and modulation frequency of 100 kHz. Spectra were recorded around the central field of 3315 G, with a sweep range of 100 G and sweep time of 20.48 s, 20 min after bringing the DPPH solution into contact with sample extracts. Spectral analysis was carried out with spectrometer build-in analysis WinEPR software (https://www.bruker.com/en/products-and-solutions/mr/epr-instruments/epr-software/winepr.html, accessed on 5 February 2025, Bruker, Billerica, MA, USA) by measuring the intensity of the central line (peak-to-peak amplitude), which is proportional to the number of spins in the sample. All of the measurements were performed in duplicate and in an analog fashion; the samples were put into the capillary which was then placed in ESR tube. The tube was immediately put in Bruker Cylindrical Super-High Q Resonator ER 4122 SHQE and measured using the given parameters. ESR results are presented in Trolox equivalents. Trolox solutions of different concentrations and samples for ESR measurements were prepared in the same way as that used for spectrophotometric determination of AC (DPPH Free Radical Assay). The AC in the samples was determined using Trolox as a standard and calculated using the following linear equation:y = 56.865x + 10.164(5)
where “y” represents the ESR signal (ARB.U) and “x” the equivalent Trolox concentration (in µM).

#### 3.4.4. Plasma Characterization by Optical Emission Spectroscopy (OES)

The experimental setup is shown in Figure 7. It consists of two parts: a reaction chamber and a spectrometer for optical emission spectroscopy (OES) measurement performed during the treatment of the samples. The reaction chamber is made of borosilicate glass and enables the monitoring of the emission in the optically visible part of the emission spectrum (from 350 nm onwards). The chamber consists of two electrodes; the upper one is grounded; the lower one has an opening for the gas inlet (argon or nitrogen) and is immersed in water. The latter is connected to a bias voltage of 20 kV at 100 Hz (Impel group d.o.o., Zagreb, Croatia). The gas flow is controlled and optimized with a flow meter (0.75 L/min).

OES of Ar and N_2_ plasmas was performed using the HR2000CG-UV-NIR spectrometer (Ocean Optics, Duiven, The Netherlands), which has a spectral resolution of 1 nm and a spectral range of 200–1100 nm. The signal was collected using a fiber (solar resistant, with 600 µm diameter and with a 5 cm lens at the end to gather more light) placed above the water surface. It thus captures all of the light emitted by the discharge sparks interacting with the water surface and the air above it. As a rule, the spectra were recorded with an integration time of 1 s and without averaging. All of the collected spectra were not corrected to a spectral response of the spectrometer, as they were only used to identify the emitted species and not for detailed findings.

#### 3.4.5. Experimental Design and Statistical Analysis

Experimental design and statistical analysis for all of the extraction parameters was performed using STATGRAPHICS Centurion (StatPoint Technologies, Inc., Warrenton, VA, USA). The experimental design included 27 US-treated samples, 36 HVED-treated samples and 9 thermally treated samples. A multilevel factorial design was used to determine the effects of the independent variables on the dependent variables. The extraction parameters had a statistically significant effect if *p* < 0.05, meaning that they were significantly different from zero at the 95.0% confidence interval.

## 4. Conclusions

The sugar beet leaf, a waste product of the sugar industry, proved to be an excellent raw material for the extraction of bioactive compounds. The US method, as one of the non-thermal extraction methods, proved to be the most suitable method for extracting the bioactive compounds and obtaining the highest AC values from the sugar beet leaves. Although in vitro methods have been successfully used to determine the AC, in vivo methods need to be performed in future research to obtain real values. Indeed, sugar beet leaves are a good source of bioactive proteins such as ribulose-1,5-bisphosphate carboxylase oxygenase enzyme (RuBisCO), whose AC was most likely not detected by spectrophotometric methods due to the nature of the compound itself (protein) and the aforementioned limitations of the DPPH and FRAP methods. For a better interpretation of the results, it is also necessary to analyze the bioavailability of the extracted bioactive compounds. Furthermore, in addition to changing the determination approach, the spectrum of the green solvents used (e.g., natural deep eutectic solvents—NADES) could also be tested in future research. In any case, the results are promising, and it is to be expected that these food wastes will increasingly be analyzed using non-thermal extraction methods.

## Figures and Tables

**Figure 1 molecules-30-00796-f001:**
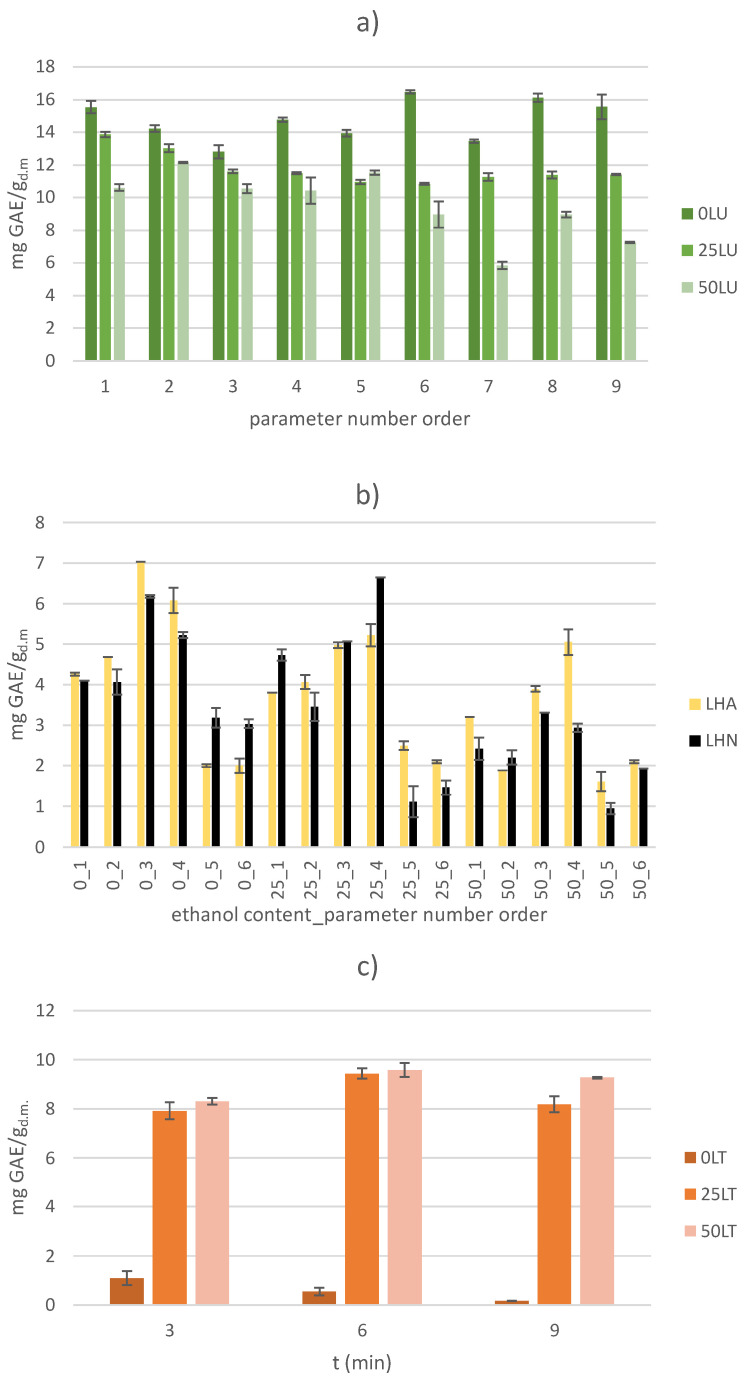
Determination of total phenolic content (TPC) for (**a**) US-treated samples, (**b**) HVED-treated samples and (**c**) thermally treated samples.

**Figure 2 molecules-30-00796-f002:**
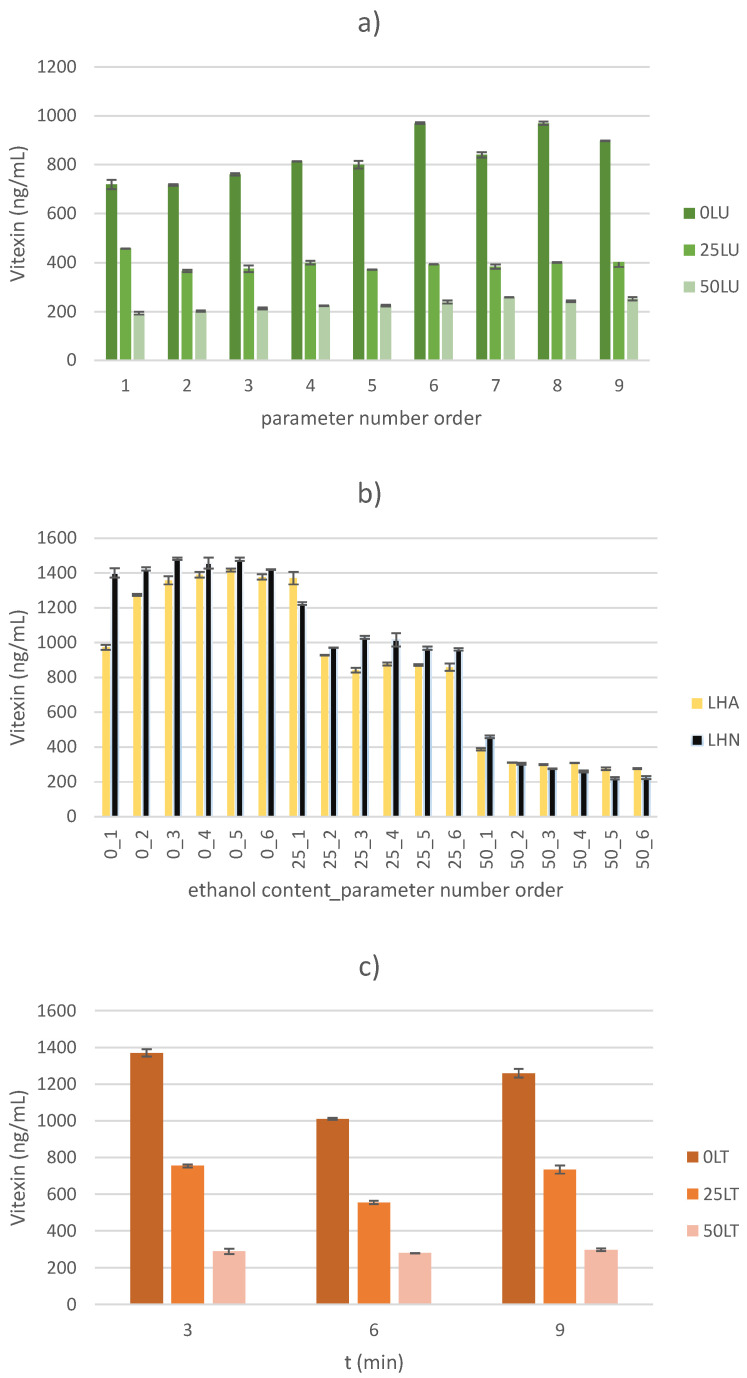
Determination of vitexin content for (**a**) US-treated samples, (**b**) HVED-treated samples and (**c**) thermally treated samples.

**Figure 3 molecules-30-00796-f003:**
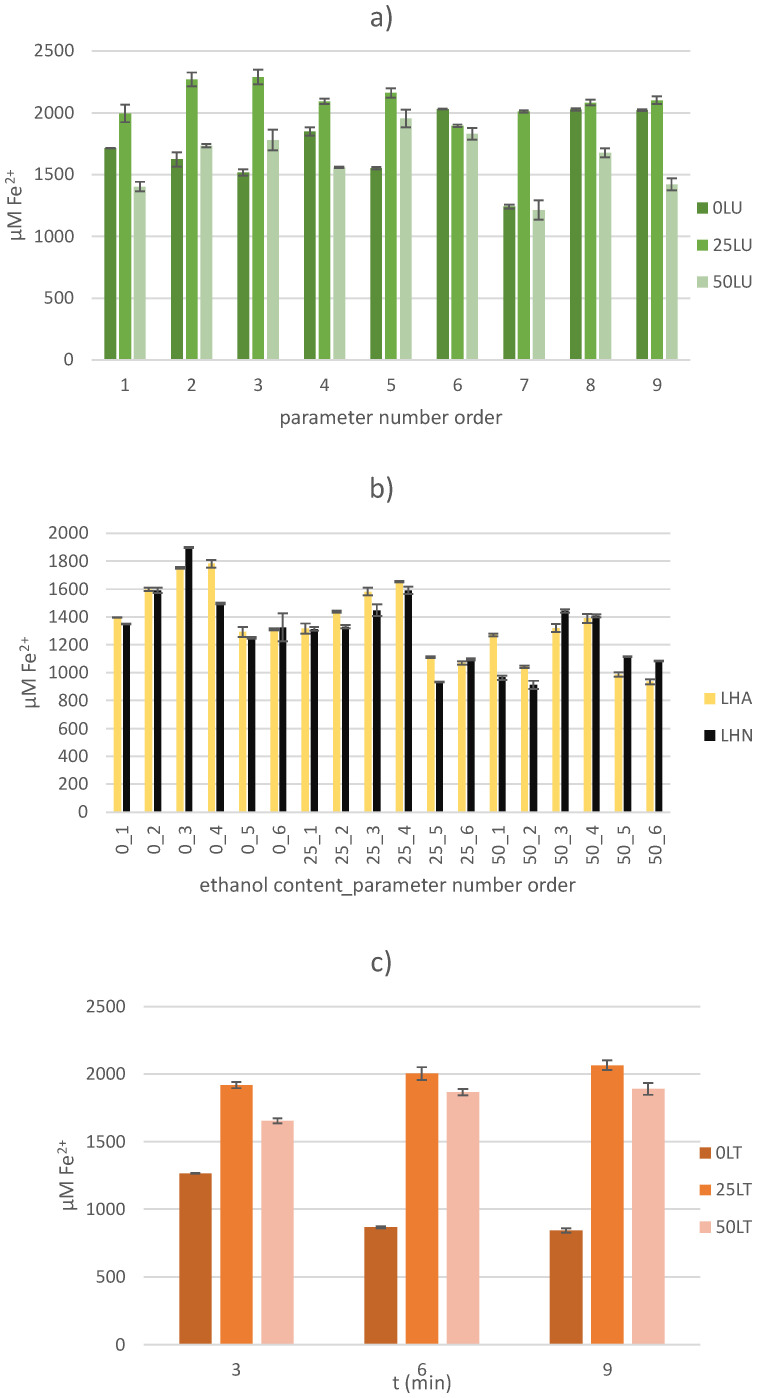
Determination of antioxidant capacity by FRAP method for (**a**) US-treated samples, (**b**) HVED-treated samples and (**c**) thermally treated samples.

**Figure 4 molecules-30-00796-f004:**
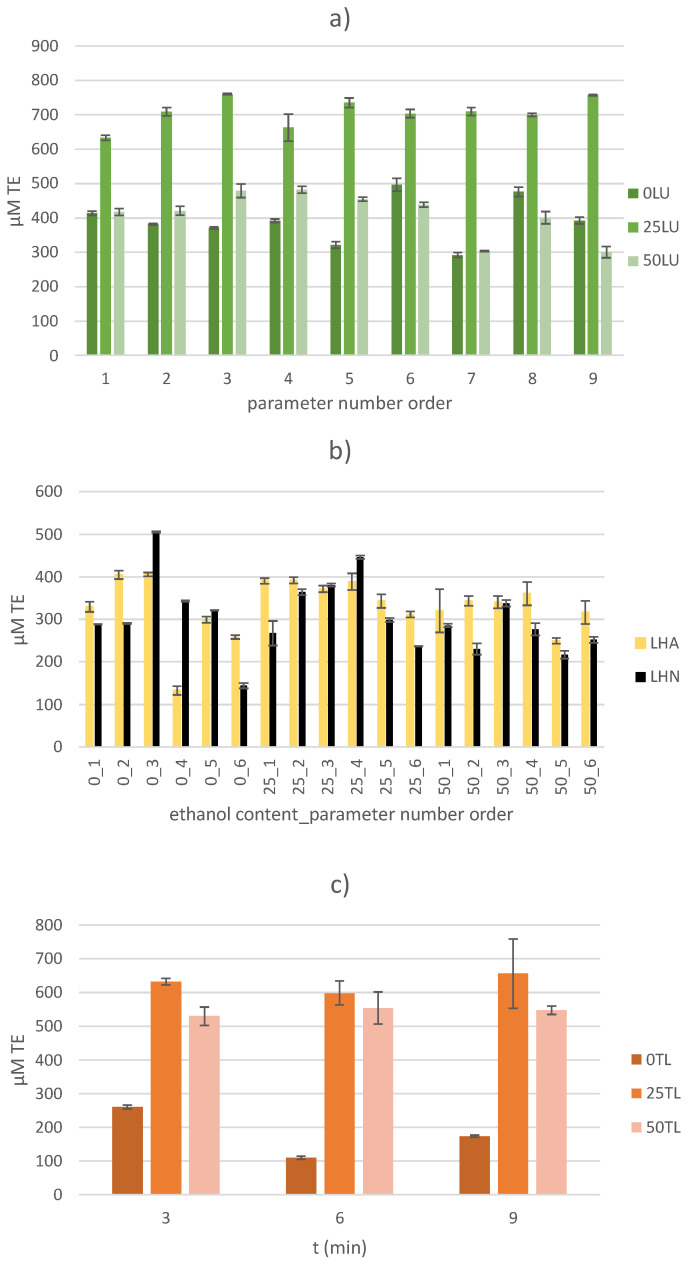
Determination of antioxidant capacity by DPPH method for (**a**) US-treated samples, (**b**) HVED-treated samples and (**c**) thermally treated samples.

**Figure 5 molecules-30-00796-f005:**
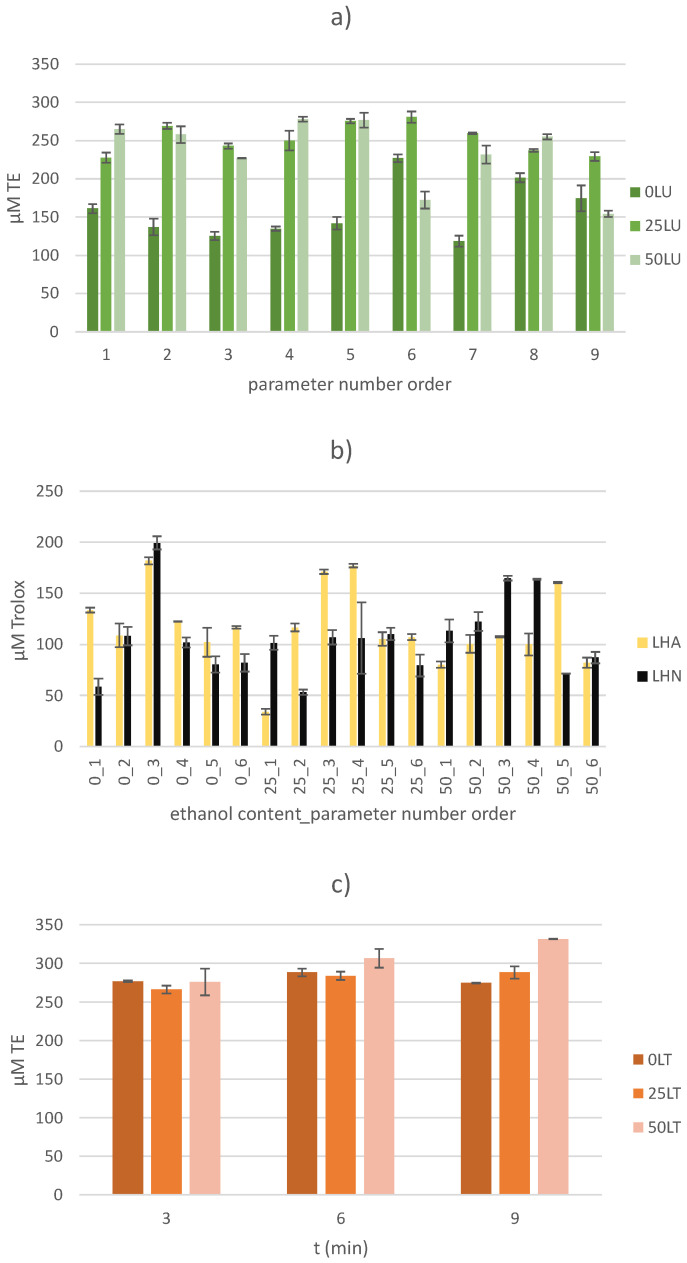
Determination of antioxidant capacity by EPR method for (**a**) US-treated samples, (**b**) HVED-treated samples and (**c**) thermally treated samples.

**Figure 6 molecules-30-00796-f006:**
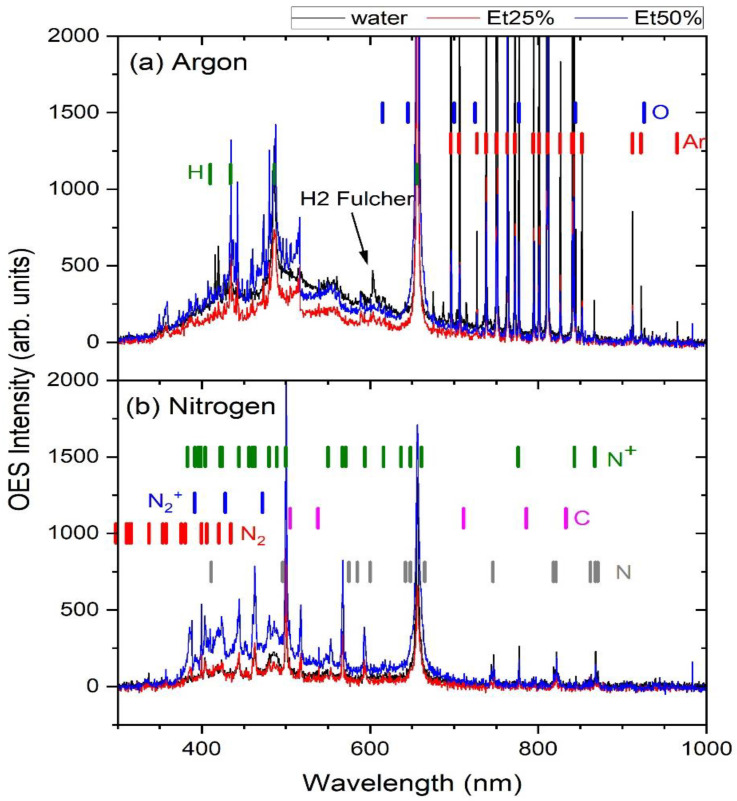
OES spectra of (**a**) argon and (**b**) nitrogen plasmas.

**Figure 7 molecules-30-00796-f007:**
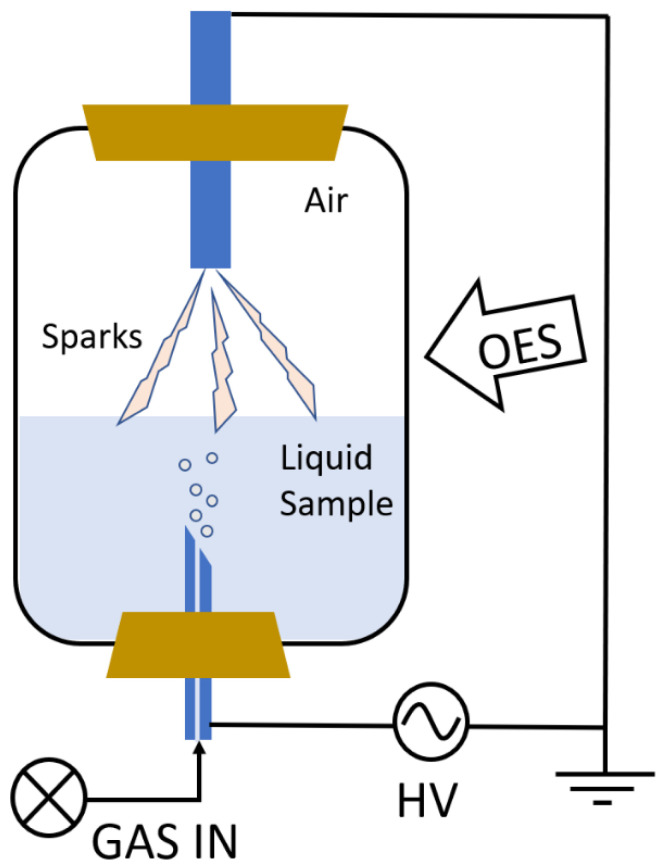
Experimental setup for plasma characterization.

**Table 1 molecules-30-00796-t001:** Sample labeling and process parameters for US-treated samples.

Ethanol Content (%)	US-Treated	Parameter Number	Amplitude (%)	Treatment Time (min)
**0** **25** **50**	**LU**	**1**	75	6
		**2**	75	3
		**3**	50	6
		**4**	50	9
		**5**	75	9
		**6**	100	9
		**7**	50	3
		**8**	100	6
		**9**	100	3

**Table 2 molecules-30-00796-t002:** Sample labeling and process parameters for HVED-treated samples.

Ethanol Content (%)	HVED-Treated	Applied Gas (A/N)	Parameter Number	Voltage (kV)	Treatment Time (min)
**0** **25** **50**	**LH**	**A** (Argon) **N** (Nitrogen)	**1**	25	6
			**2**	20	6
			**3**	20	9
			**4**	25	9
			**5**	20	3
			**6**	25	3

**Table 3 molecules-30-00796-t003:** Sample labeling and process parameters for thermally treated samples.

Sample Name	Ethanol Content (%)	Treatment Time (min)
**0LT3**	0	3
**0LT6**	0	6
**0LT9**	0	9
**25LT3**	25	3
**25LT6**	3	6
**25LT9**	25	9
**50LT3**	50	3
**50LT6**	50	6
**50LT9**	50	9

## Data Availability

Data is contained within the article.

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
