# Peer review of "High-Power Ultrasound and High-Voltage Electrical Discharge-Assisted Extractions of Bioactive Compounds from Sugar Beet (Beta vulgaris L.) Waste: Electron Spin Resonance and Optical Emission Spectroscopy Analysis"

_molecules, 2025, doi:10.3390/molecules30040796_

Round 1

Reviewer 1 Report

Comments and Suggestions for Authors

In general, this is nice work.

However, I have one Major objection to be considered and explained:

Since authors clearly marked vitexin as the most dominant phenolic in your material and since it is flavonoid I am wondering why authors did not consider to measure and monitor Total flavonoid content (TFC) gather with TPC? This assay is much more specific and precise unlike to TPC which has a lot of intereferences which can lead to misleading results.

Technical suggestion

Please check your references. I think they not ordered in an appropriate way. Some of them are given by name of authors instead of numbers. For instance, in the Introduction section There is reference no. 1 and than next are 7,8 and 9. Between there are two references by names. Similar error is repeating also later in text. Please check all and update to be ordered correctly.

Minor corrections

Lines 60-65: Please provide reference(s) for given statement(s).

Lines 78-81: Same as previous.

Line 97: Here also should be specified that apart from presence, for biological activity of flavonoids position and number of OH group(s) is also extremely important.

Line 111: It should be "chlorophylls" and "carotenes" in plural.

Lines 111 and 112: To be precise, phytochemicals ARE secondary metabolites. Please revise.

Line 129: Strongly suggest to rewrite "... of the results from before..." as follow "...of the previously published results....".

Lines 222-223 and 228-229: Authors stated (Lines 222-223) that vitexin is not soluble in water in EtOH, however, your later Discussion (Lines 228-229) suggests opposite? Please check/clarify/revise.

Line 277: typo- put Latin name in Italic here.

Line 447: "DPPH radical". Correct.

Kind regards.

Author Response

Response to Reviewer 1 Comments

Introductory acknowledgement to the reviewer

Dear reviewer,

at the beginning we would like to thank You for the excellent review. We believe that our work has been visibly improved, precisely because of Your excellent suggestions.

Thank You once again!

Sincerely Yours,

Authors

Comment 1:

In general, this is nice work.

However, I have one Major objection to be considered and explained:

Since authors clearly marked vitexin as the most dominant phenolic in your material and since it is flavonoid I am wondering why authors did not consider to measure and monitor Total flavonoid content (TFC) gather with TPC? This assay is much more specific and precise unlike to TPC which has a lot of intereferences which can lead to misleading results.

Response 1:

Dear reviewer, first of all, thank you very much for your kind words. We are pleased that you liked the work. Unfortunately, as you wrote yourself, TPC is less specific and besides flavonoids, the concentration of some other compounds (e.g. phenolic acids) also influences the result. Although TFC is a more specific method, the authors opted for an even more detailed and validated HPLC-MS/MS method. After reviewing the available literature (Maravić et al., 2021*), vitexin was found to be the most abundant sugar beet flavonoid, and instead of the simpler and less specific TFC method, the authors opted for a much more specific and precise method and determined vitexin using the HPLC-MS/MS method.

* - https://www.sciencedirect.com/science/article/abs/pii/S2352554122001322?via%3Dihub

Comment 2:

Technical suggestion

Please check your references. I think they not ordered in an appropriate way. Some of them are given by name of authors instead of numbers. For instance, in the Introduction section There is reference no. 1 and than next are 7,8 and 9. Between there are two references by names. Similar error is repeating also later in text. Please check all and update to be ordered correctly.

Response 2:

We thank you for your attention and apologize for the omission. The references have been revised according to your instructions.

Comment 3:

Minor corrections

Lines 60-65: Please provide reference(s) for given statement(s).

Response 3:

Thank you for the suggestion. It has been accepted and references 7 and 8 have been added to the manuscript.

Comment 4:

Lines 78-81: Same as previous.

Response 4:

Thank you for the suggestion. It has been accepted and references 10 has been added to the manuscript.

Comment 5:

Line 97: Here also should be specified that apart from presence, for biological activity of flavonoids position and number of OH group(s) is also extremely important.

Response 5:

Thank you for this excellent suggestion. The sentence "In addition to the presence, the position and number of OH groups are also of great importance for the biological activity of flavonoids", together with reference no. 14, which supports the above, has been included in the manuscript.

Comment 6:

Line 111: It should be "chlorophylls" and "carotenes" in plural.

Response 6:

Thank you for your comment, the above words have been corrected in the manuscript according to your instructions.

Comment 7:

Lines 111 and 112: To be precise, phytochemicals ARE secondary metabolites. Please revise.

Response 7:

We apologize for the oversight. Instead of "and", a "-" had to be placed between phytochemicals and secondary metabolites. In accordance with your instructions, the sentence has been reworded in the manuscript.

Comment 8:

Line 129: Strongly suggest to rewrite "... of the results from before..." as follow "...of the previously published results....".

Response 8:

Thank you for the suggestion. It has been included in the manuscript.

Comment 9:

Lines 222-223 and 228-229: Authors stated (Lines 222-223) that vitexin is not soluble in water in EtOH, however, your later Discussion (Lines 228-229) suggests opposite? Please check/clarify/revise.

Response 9:

Thank you for your excellent observation and we apologize for the confusion. The authors intended to write that the solubility of vitexin in solvents such as water and ethanol is significantly lower compared to some organic solvents such as DMSO. This has been corrected in the manuscript to avoid confusion for future readers.

Comment 10:

Line 277: typo- put Latin name in Italic here.

Response 10:

We apologize for the oversight. In accordance with your instructions, it has been revised as requested.

Comment 11:

Line 447: "DPPH radical". Correct.

Response 11:

We apologize for the oversight. In accordance with your instructions, it has been revised as requested.

Reviewer 2 Report

Comments and Suggestions for Authors

The manuscript entitled “High-power ultrasound and high voltage electrical discharge assisted extractions of bioactive compounds from Sugar beet (Beta vulgaris L.) waste: electron spin resonance and optical emission spectroscopy analysis” is well-written, well-structured, well-designed and interesting due to the extraction methods used and also due to the fact that waste-type plant material is used.

I would like to make a few observations:

1. The introduction is a bit too long. The first 2 paragraphs could be shortened. If you could define "plasma".

2. What other polyphenolic components have been identified and quantified in Beta vulgaris leaf extracts? (from the literature).

3. Do the authors consider that vitexin is the major component in the extracts obtained and studied by them?

4. Did you use 25 and 50% ethanol. On what basis? 70% ethanol was not syndicated for the extraction of polyphenols?

5. The results obtained in the determination of total polyphenols were expressed in mg/g, and for vitexin in ng/ml.

Author Response

Response to Reviewer 2 Comments

Introductory acknowledgement to the reviewer

Dear reviewer,

at the beginning we would like to thank You for the excellent review. We believe that our work has been visibly improved, precisely because of Your excellent suggestions.

Thank You once again!

Sincerely Yours,

Authors

Comment 1:

Comments and Suggestions for Authors

The manuscript entitled “High-power ultrasound and high voltage electrical discharge assisted extractions of bioactive compounds from Sugar beet (Beta vulgaris L.) waste: electron spin resonance and optical emission spectroscopy analysis” is well-written, well-structured, well-designed and interesting due to the extraction methods used and also due to the fact that waste-type plant material is used.

I would like to make a few observations:

1.The introduction is a bit too long. The first 2 paragraphs could be shortened. If you could define "plasma".

Response 1:

Dear reviewer, first of all, thank you sincerely for your thoughtful feedback. We really appreciate your kind words and are pleased that you found our work useful.

Thank you for your commendable comment. In the interest of a better understanding of the topic and a better reading flow, the authors believe that shortening the slightly longer paragraphs would negatively affect the reading flow. However, an explanation of what plasma is (with reference) has been included in the manuscript and states as follows: “ Plasma is one of the four fundamental states of matter, i.e. a fully or partially ionized gas consisting of cations, anions, radicals, electrons and excited and unexcited atoms [10].“

Comment 2:

  1. What other polyphenolic components have been identified and quantified in Beta vulgaris leaf extracts? (from the literature).

Response 2:

Thank you for your inquiry. There are numerous studies on sugar beet pulp as well as on other types of beet, but very few specifically on sugar beet leaves. Nevertheless, in the introduction of the manuscript, in lines 114 to 118, you will find polyphenolic compounds of sugar beet leaves: „There are relatively few studies on the extraction and characterization of the bioactive components contained in sugar beet leaves but in general sugar beet leaves contain high amounts of gallic acid (344 µg/mL) and ferulic acid (89.7 µg/mL), while cinnamic and caffeic acids and catechin are present in small amounts [21]. In addition, vitexin was found to be the most abundant phenolic compound [22].“

Comment 3:

  1. Do the authors consider that vitexin is the major component in the extracts obtained and studied by them?

Response 3:

Thank you for your comment. According to previous studies, proteins are one of the most common components of sugar beet leaves. However, in this work, the focus is on phenolic compounds, and considering the available literature (Maravić et al., 2022*), the flavonoid vitexin is the most abundant. For that reason, vitexin was more precisely determined and quantified using the HPLC-MS/MS method.

* - https://www.sciencedirect.com/science/article/abs/pii/S2352554122001322?via%3Dihub

Comment 4:

  1. Did you use 25 and 50% ethanol. On what basis? 70% ethanol was not syndicated for the extraction of polyphenols?

Response 4:

Thank you for your inquiry. The solvents used in this study were: water, 25 and 50% aqueous ethanol solutions. A statistical analysis of the preliminary results showed a decrease in TPC values when higher ethanol concentrations were used in US and HVED-treated samples (non-thermal and sustainable extraction methods). For the above reasons, extractions with ethanol concentrations higher than 50% were not justified.

Comment 5:

  1. The results obtained in the determination of total polyphenols were expressed in mg/g, and for vitexin in ng/ml.

Response 5:

Thank you for taking note. For better comparability with the results from the previous article, the TPC results are given in mg/g. In addition, the mentioned method has not been validated. The method for the determination of vitexin, on the other hand, is a validated method and the results should be reported according to the validation, which in this case is ng/mL.
